# Bayesian Network Analysis of Lysine Biosynthesis Pathway in Rice

Aditya Lahiri [1,*,†] , Khushboo Rastogi [2,3] , Aniruddha Datta [1] and Endang M. Septiningsih [2,3]

1   Department of Electrical and Computer Engineering, Texas A&M University, College Station, TX 77843, USA; datta@ece.tamu.edu
2   Department of Soil and Crop Sciences, Texas A&M University, College Station, TX 77843, USA; khushboorastogi5@tamu.edu (K.R.); eseptiningsih@tamu.edu (E.M.S.)
3   Genetics Interdisciplinary Program, Texas A&M University, College Station, TX 77843, USA
*   Correspondence: adi441994@gmail.com; Tel.: +1-832-782-4580

**Abstract:** Lysine is the first limiting essential amino acid in rice because it is present in the lowest quantity compared to all the other amino acids. Amino acids are the building block of proteins and play an essential role in maintaining the human body's healthy functioning. Rice is a staple food for more than half of the global population; thus, increasing the lysine content in rice will help improve global health. In this paper, we studied the lysine biosynthesis pathway in rice (*Oryza sativa*) to identify the regulators of the lysine reporter gene *LYSA* (LOC_Os02g24354). Genetically intervening at the regulators has the potential to increase the overall lysine content in rice. We modeled the lysine biosynthesis pathway in rice seedlings under normal and saline (NaCl) stress conditions using Bayesian networks. We estimated the model parameters using experimental data and identified the gene *DAPF*(LOC_Os12g37960) as a positive regulator of the lysine reporter gene *LYSA* under both normal and saline stress conditions. Based on this analysis, we conclude that the gene *DAPF* is a potent candidate for genetic intervention. Upregulating *DAPF* using methods such as CRISPR-Cas9 gene editing strategy has the potential to upregulate the lysine reporter gene *LYSA* and increase the overall lysine content in rice.

**Keywords:** lysine; rice; amino acids; saline stress; abiotic stress; gene regulatory network; Bayesian network; parameter estimation; inference; RNA Seq

## 1. Introduction

### 1.1. Background

Proteins are one of the primary building blocks of all life on Earth and are present in every cell in the human body. Proteins are a crucial macronutrient in the human diet; they help build and repair cells and are essential for the human body's growth and development [1]. Proteins are comprised of long chains of amino acids; once the human body digests the proteins, they are broken down into their constituent amino acids [2]. There are twenty naturally existing amino acids that encode the 20,000 (approximate) unique proteins in the human body [3]. Among these amino acids, nine are classified as essential, and eleven are classified as nonessential [2,3]. Amino acids produced by the human body are considered nonessential, whereas the amino acids that cannot be synthesized by the body are considered essential [3]. Essential amino acids include phenylalanine, valine, tryptophan, threonine, isoleucine, methionine, histidine, leucine, and lysine [4]. Since essential amino acids cannot be synthesized, they need to be introduced to the human body through diets rich in complete proteins. A protein food source is considered a complete protein if it contains all the essential amino acids [5]. Typically, animal-based proteins are considered sources of complete protein. Plant-based proteins are considered incomplete as they do not contain all the essential amino acids [5,6].

According to the National Academy of Medicine, the recommended dietary allowance (RDA) of protein intake is 0.8 g/kg/day [7,8]. A diet deficient in protein can cause edema, thinning of hair, and muscle mass loss in adults [9]. Though protein deficiency is rare in the developed world, it is still prevalent in impoverished and underdeveloped countries, especially among children [9,10]. Plant-based proteins accounted for 57% of the global protein supply and were followed by animal-based proteins such as meat and dairy, which accounted for 18% and 10%, respectively [11]. Even though plant-based proteins constitute a majority of the global protein supply, according to the World Health Organization (WHO), the demand for animal-based protein has been on the rise due to urbanization, population growth, and rising economies. The WHO predicts that annual meat production will reach 376 million tons by 2030, a 72% increase from 1997–1999, when the yearly meat production was 218 million tons [12]. This global increase has placed a burden on the livestock sector, especially in Europe and the Americas, where animal-based protein intake is higher than that of plant-based proteins [13]. In the USA and European countries, proteins from animal-based sources ranged from 55% to 71% (depending on countries) of the total protein intake, a significant proportion of which were from red meat [14].

Animal-based protein sources such as meat, milk, and eggs are richer in essential amino acids and have a higher food protein quality in terms of digestibility, net protein utilization, and biological value compared to plant-based protein sources such as legumes and cereals [13]. However, animal-based proteins, specifically processed and red meats, have been linked with cancer, type 2 diabetes, and cardiovascular diseases [15–17]. Apart from health concerns, proteins sourced from animals have a significant impact on climate change. According to the Food and Agriculture Organization of the United Nations, the livestock supply chain accounts for 14.5% of global anthropogenic greenhouse gas emissions [18]. With the global population set to reach 9.8 billion by 2050 and the increasing demand for animal-based proteins, the challenges associated with food security and climate change will only be exacerbated [12,19]. Hence, a shift toward plant-based protein sources may help reduce the carbon footprint, risks of chronic illness, and food security. While plant-based proteins may not contain all the necessary essential amino acids, a diet containing a diverse range of plant proteins can help overcome this limitation [20]. Cereal plants such as wheat, rice, and maize constitute the primary protein sources in developing countries [21,22]. With the majority of the world's population living in developing countries, it will therefore be beneficial to increase the protein content in cereal plants to ensure food security and prevent malnutrition.

### 1.2. Lysine Content in Rice

Lysine is produced in the aspartate pathway along with three other essential amino acids: threonine, methionine, and isoleucine [23]. Lysine is also the first limiting essential amino acid in cereal and legume crops because it is present in the lowest quantity [23–25]. This is why lysine deficiency is a common problem in developing nations that rely heavily on cereal crops [23,26]. A lysine deficient diet can reduce immunity, decrease protein levels in the blood, and cause retardation of mental and physical development in children [24]. Rice is a cereal plant that is an important food source for more than 50% of the global population [27]. About 95% of global rice is produced in developing countries, among which 92% are countries in Asia [28]. Rice accounts for 50% of the dietary caloric supply for 520 million living in poverty in Asia [29]. Like most cereal crops, rice is deficient in lysine, so in this study, we are interested in identifying the genetic regulators of lysine production in rice, since intervening at these regulators has the potential to increase the free lysine content in rice grains [30]. Enriching lysine content in rice will be a step toward ensuring food security and preventing malnutrition, especially in the vulnerable segments of the global population.

Over the last 50 years, lysine metabolism has been extensively studied. It has been shown that lysine is a self-regulating amino acid as the lysine biosynthesis pathway has two inhibition feedback loops [31–33]. These feedback loops are activated by the free lysine content, which negatively regulates the enzymes dihydrodipicolinate synthase (DHPS) and

aspartate kinase (AK) [24,34]. AK is the first enzyme of the lysine biosynthesis pathway and is also inhibited by threonine, another essential amino acid synthesized by the aspartate pathway [24,34]. Lysine is also degraded through the enzymes lysine ketoglutarate reductase (LKR) and saccharopine dehydrogenase (SDH) bifunctional enzymes [34]. The LKR and SDH enzymes are present in the saccharopine pathway and they initiate the lysine catabolism process through the TCA cycle (tricarboxylic acid cycle) [24]. The metabolic pathway of lysine biosynthesis and catabolism in presented in Figure 1 [31–33]. Thus, lysine can be enriched in cereal plants by enhancing its production in the biosynthesis pathway, preventing its catabolism, or combining these two approaches. A study by Long et al. (2013) focused on enhancing lysine through metabolic engineering of rice. These transgenic lines of rice overexpressed AK and DHPS. They observed that LKR and SDH levels were significantly higher in seeds of these rice lines, implying that the catabolic enzymes LKR and SDH were counteracting the effects of transgene AK and DHPS [33]. This method increased the free lysine content by 1.1 times in transgenic lines compared to the wild type. This study also implemented an LKR-RNAi line, which showed a 10-fold increase in lysine content, and a combination of LKR-RNAi with AK/DHPS overexpressing lines led to a 60-fold increase in free lysine content. In a different study, Yang et al. (2016) developed two pyramid transgenic lines in rice. The lysine content in these transgenic lines showed increased lysine content up to 25-fold. This was achieved by enhancing the biosynthesis pathway and suppressing the catabolism pathway at the same time [35]. Unlike many lysine enhancement studies, which lead to reduced yield, oil content, and phenotype change, no significant trait changes were observed in this case, and the developed transgenic rice was deemed favorable for commercialization [36–38].

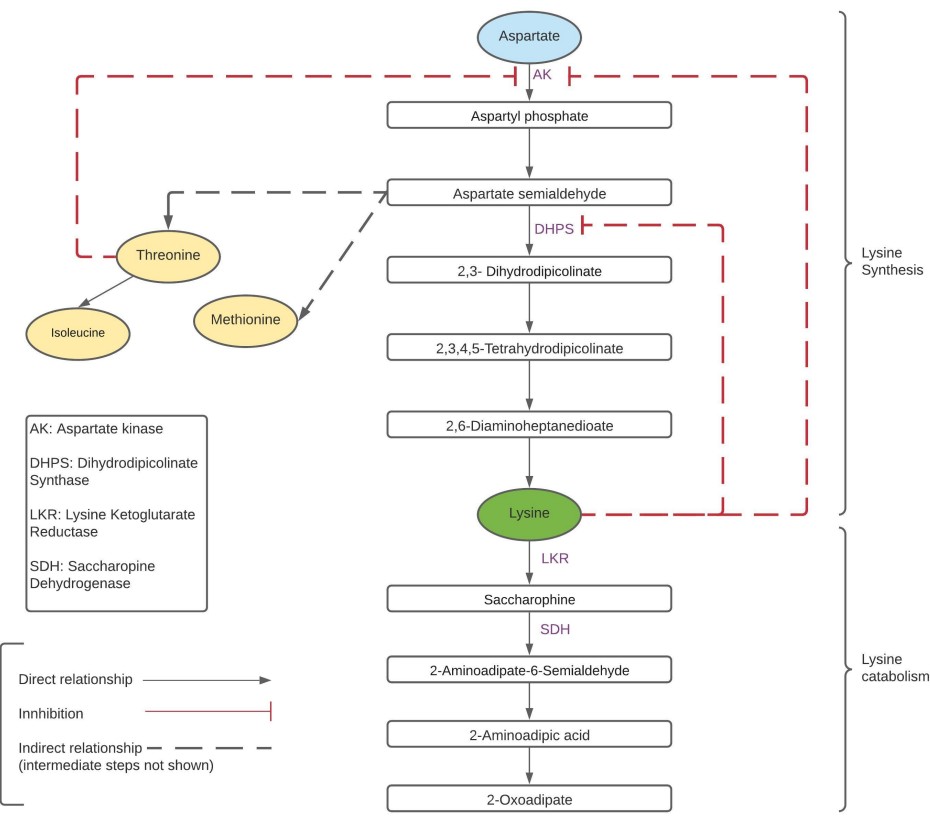

**Figure 1.** Lysine metabolic pathway for synthesis and catabolism.

While these studies have demonstrated that lysine content can be enhanced through careful metabolic engineering of high-lysine transgenic lines, these are not yet commercialized. Furthermore, transgenic crops rely on introducing foreign genes (transgenes) into

the host crop, making them vulnerable to public acceptance. That is why, in this paper, we are interested in understanding the underlying genetic regulatory networks (GRNs) that govern these complex interactions. The GRNs can help us identify the genetic regulators of lysine that later on can be targeted using gene editing methods such as CRISPR-Cas9. Unlike transgenic crops, the final product of gene editing can be cleared of any foreign DNA segments. Instead of relying on transgenic insertions, gene editing may knock out or replace targeted native genes in the genome of the crop to give rise to desirable traits. The United States Department of Agriculture (USDA) has allowed gene edited crops to be labeled as non-GMO, which will make gene edited crops significantly less controversial than transgenic crops [39]. A recent study by Shew et al. showed that gene edited crops were preferred over GMO crops in multiple countries [40]. Thus, by studying the underlying GRN involved in lysine regulation in rice, we can identify potential targets for gene editing.

LKR and SDH are known regulators of lysine in the catabolic pathway, and genetically intervening them have been proven to prevent lysine degradation [41]. Therefore, in this paper, we focus on identifying lysine regulators in the biosynthesis pathway. Overexpressing the regulators in the biosynthesis pathway through gene editing techniques such as CRISPR-Cas9 has the potential to increase the free lysine content in rice. In Figure 2, we derive the GRN of the lysine biosynthesis pathway in rice (*Oryza sativa*) from the KEGG pathways database [42]. Each rectangular box in Figure 2 represents a gene in the lysine biosynthesis pathway. The gene names are annotated according to their respective MSU IDs (LOC_Os##g#####) [43]. In addition to the MSU IDs, the boxes contain alphabets in red font within parenthesis. These alphabets are used as an alias for genes in the later sections of the paper. Genes I–N have been given names in the literature and these names have been mentioned in the boxes alongside their MSU IDs, e.g., gene K (LOC_Os03g09910) is also known as *ALD1*. The genetic interactions converge at *LYSA* (LOC_Os02g24354 or gene N), which positively regulates the amino acid lysine (L-Lysine, where the $\alpha$ carbon is in the S configuration ). This makes *LYSA* (gene N) a reporter gene of lysine. Thus, our objective is to identify genes that will upregulate *LYSA*.

To identify the *LYSA* regulators, we will model the GRN of the lysine biosynthesis pathway using Bayesian networks (BN). We will then use publicly available data to infer the BN model's parameters. The model can then be used to identify the genes that upregulate *LYSA*. This modeling pipeline is similar to our previous work where we identified regulators of drought response in *Arabidopsis* [44,45]. We identify the *LYSA* regulators under normal and saline stress (NaCl) conditions. Soil salinity is one of the significant environmental constraints on the crop life cycle. Nearly 5% (77 million hectares) of the global arable land has excess salinity [46]. Due to various factors such as climate change and irrigation malpractices, the soil salinity is predicted to increase by 16.2 million hectares by 2050 [47,48]. Among abiotic stresses, soil salinity is the second largest cause of crop loss in rice after drought [49,50]. Saline stress primarily affects rice during its seedling, early vegetative, and reproductive stages [49,51]. We have extensively studied and identified regulators of drought response in our previous work [44,45]; in our current study however, we focus on saline stress in rice. We are specifically interested in observing if the *LYSA* regulators change under saline stress. Stewart et al. showed that saline stress leads to the accumulation of aspartic acid (aspartate), which is the first element in the lysine biosynthesis pathway [52]. Furthermore, it has been reported that under stressed conditions, aspartic acid catabolizes into asparagines, threonine, lysine, isoleucine, and methionine [53]. Studies involving maize and wheat showed increased lysine content under saline stress; however, the precise effect of saline stress on the lysine content in rice remains to be explored [54,55].

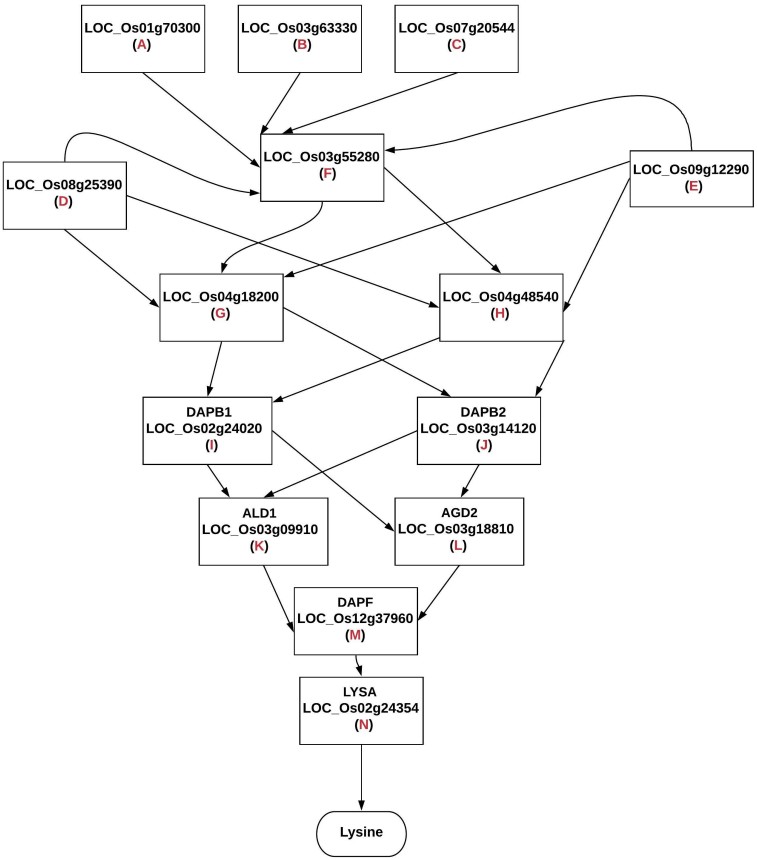

**Figure 2.** Gene regulatory network for lysine biosynthesis pathway in rice. The gene names are presented according to their MSU IDs. The alphabets in red font are aliases for the respective genes, e.g., LOC_Os01g70300 is referred to as gene A. Genes I–N have been given names in the literature; these have been mentioned in the figure alongside their respective MSU IDs.

## 2. Materials and Methods

GRNs describe the complex interactions taking place between regulators and their target genes. Typically, regulators consist of transcription factors (TFs), genes, RNA binding proteins, and regulator RNAs that can control the gene expression of the target genes [56–58]. GRNs govern the decision-making process in response to endogenous and external stimuli; thus, understanding their behavior at the genomic level can give us critical insights into achieving desirable phenotypical traits like increased lysine content [59,60]. GRNs have been modeled extensively in the past for a wide range of applications such as discovering novel biological relationships, studying complex diseases, drug design, and developing pathogen-resistant crops [61–65]. Common modeling techniques include differential equations, linear models, Boolean networks, probabilistic Boolean networks, Bayesian networks, and small molecule level models [66–70]. Each technique has its set of advantages and limitations. Therefore, we must consider the nature of the interactions in the GRN and the overall domain of the study while selecting a modeling method. In this paper, we are interested in modeling the lysine biosynthesis pathway in rice under normal (unstressed) and saline stress conditions. The interactions taking place in the pathway are sparse, multivariate, and stochastic in nature. Furthermore, with the advent of high-throughput technologies, publicly available genomic data have become easily accessible [71]. Due to these factors, we will model the lysine biosynthesis pathway using Bayesian networks (BNs). BNs provide a stochastic framework and allow integration of pathway knowledge and data.

*2.1. Bayesian Network Modeling*

BNs are a class of probabilistic graphical models (PGM) that integrate probability and graph theory to represent stochastic and causal relationships among variables in a system [72,73]. BNs consist of two main components (i) a directed acyclic graph (DAG) and (ii) local probability distributions (LPD) or the network parameters [74]. The DAG is a map that describes the causal relationships among the system variables, also known as nodes. DAGs specify the dependencies among the nodes and explain the flow of cause and effect in the overall network. The DAG can be derived from the literature or estimated from data using structure learning algorithms [75]. Associated with each node in the DAG is a local probability distribution (LPD) that describes the stochastic nature of interaction among the connected nodes [73]. The LPDs and the DAGs together describe the factorization of the joint probability distribution of all the nodes in terms of their LPDs. In order to formalize this notion, consider a BN with $N$ nodes such that it has a DAG structure $\mathcal{G}(X, E)$, where $X_i$ represents the $i$th node in the set of nodes $X$ and $E$ represents the set of casual edges between the nodes. Now, suppose the LPD for each node $X_i$ is given by $P(X_i \mid P_a(X_i))$, where $P_a(X_i)$ is the set of parent nodes of $X_i$. Then, by the local Markov independence assumption, each node, given its parent nodes, is independent of its nondescendant nodes. We can then factorize the joint probability of all the nodes in $X$ as:

$$P(X = \{X_1, X_2, ..., X_i, ..., X_N\}) = \prod_{i=1}^{N} P(X_i | P_a(X_i)) \tag{1}$$

To model the lysine biosynthesis pathway using BN, we construct a DAG from the Kegg pathway that we discussed in Figure 2. Learning the DAG from data is an NP-hard problem and often requires selecting a graph structure from a candidate of possible DAGs [76,77]. This is a computationally expensive task, and the size of publicly available genomic data sets is not sufficiently large to produce a reliable DAG. Therefore, we use pathway information (see Figure 2) to construct the DAG for the lysine biosynthesis pathway in Figure 3. Every node (represented by circles) in the DAG represents a gene present in the lysine biosynthesis pathway. These genes are referenced by their aliases; for instance, gene N represents *LYSA*. The nodes are connected by arrows that represent actual biological relationships as described in the pathway. We assume that genes in the network can be active, dormant, or inhibited. Thus, we model each node as a categorical random variable with three states 1 (active), 0 (dormant), and −1 (inhibited). Associated with each node is a rectangular box that describes the LPD (network parameter). For Node A, $\theta_A$ is a vector representing the marginal probability of gene A being active, dormant, or inhibited. Similarly, $\theta_{M|L,K}$ is a vector representing the conditional probability of gene M being active, dormant, or inhibited given the states of its parents, gene L and gene K. This completes our discussion of the DAG for the lysine biosynthesis pathway. In the next section, we will discuss how to estimate the LPDs. Once all the LPDs have been calculated, the Bayesian network model is complete and can be used to perform gene intervention simulation under normal and saline stress conditions. These simulations will help us gain insight into the effect of intervening at the various genes. Genes that upregulate *LYSA* (gene N) will be considered ideal targets for genetic intervention. Interventions in the GRN can be carried out using gene editing methods such as CRISPR-Cas9 [60]. A simple example BN with its LPDs is shown in Section 2.3 for the purpose of demonstrating inference in BN. This example might be useful in developing a better understanding of the DAG structure and the LPDs.

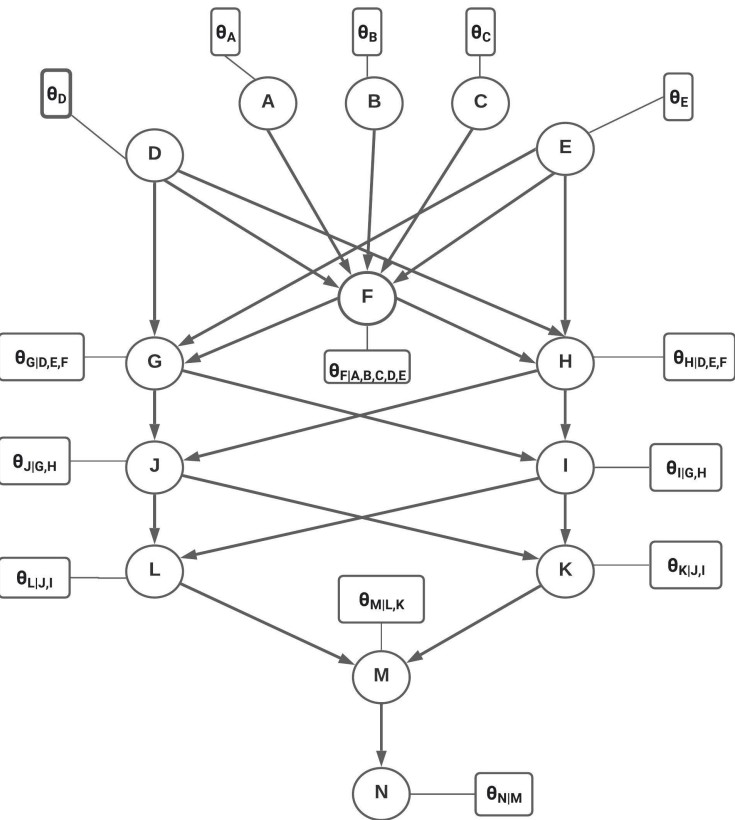

**Figure 3.** Directed acyclic graph (DAG) of the lysine biosynthesis pathway. Each node (circle) represents a gene in the pathway. The rectangular boxes represent the local probability distributions of the respective nodes. Each node is modeled as a categorical random variable with the following states: active (1), dormant (0), and inhibited ($-1$).

## 2.2. Parameter Estimation

Several methods can be employed to estimate the LPDs (network parameters) in a BN. Frequentist approaches such as maximum likelihood estimation (MLE) are common when estimating the LPDs in a BN [78]. However, we will use a Bayesian approach to estimate the LPDs for the DAG constructed in the previous section. This is because the sizes of publicly available data sets are not sufficiently large to be reliably used by data-driven frequentist approaches. Unlike frequentist approaches, Bayesian estimation produces a posterior probability distribution for the LPDs based on data and prior knowledge [79]. The point estimate for the LPDs can be obtained by approximating the posterior distributions by their expected value or mode [80]. The Bayesian estimation process is based on Bayes's rule where the posterior distribution of a random variable $X$, for a data set $\mathcal{D}$, is given by:

$$P(X|D) = \frac{P(\mathcal{D}|X)P(X)}{P(\mathcal{D})} \tag{2}$$

where P(X) is the prior distribution of X.

We will now use this approach to derive the general expression for estimating the LPDs for a BN where the nodes are modeled as categorical random variables. We can then extend our findings to the DAG in Figure 3.

Consider a BN with a DAG denoted by $\mathcal{G}$ containing $N$ ($N$ is a natural number) nodes. Each node $X_i$ in $\mathcal{G}$ is modeled as a categorical random variable with the following states: active (1), dormant (0), and inhibited (0). Thus, for any node $X_i$ in $\mathcal{G}$, $X_i \in \mathbf{S} = \{1, 0, -1\}$, so if $X_i = 0$, it implies that the node $X_i$ is dormant. Let the probability with which $X_i$ assumes

any of the states in set **S** be given by the probability vector $\theta_{X_i}$. Then, $\theta_{X_i}$ is of the form $[\theta_{X_i=1}, \theta_{X_i=0}, \theta_{X_i=-1}]^T$, where $\theta_{X_i=s}$ represents the probability of $X_i=s$ for s $\in$ **S** and $\Sigma_s \theta_{X_i=s}$ = 1. Now, suppose we have a data set $\mathcal{D}$ that contains $n$ ($n$ is natural number) independent and identically distributed (i.i.d.) observations for each of the $N$ nodes in $\mathcal{G}$. For a node $X_i$ in $\mathcal{G}$, let $M_{X_i}[\mathbf{S} = \mathrm{s}]$ represent the frequency of $X_i = \mathrm{s}$ in $\mathcal{D}$ ($\Sigma_s M_{X_i}[\mathrm{s}] = \mathrm{n}$). Then, the likelihood under the data set $\mathcal{D}$ can be modeled as:

$$P(X_i|P_a(X_i), \theta_{X_i}) \sim Multinomial(\theta_{X_i}, n) \tag{3}$$

$$Multinomial(\theta_{X_i}, n) = n! \prod_{s \in \mathbf{S}} \frac{\theta_{X_i}^{M_{X_i}[s]}}{M_{X_i}[s]!} \tag{4}$$

The Bayesian estimation process requires selecting a prior distribution. Prior distributions can be selected based on domain knowledge; however, in the absence of domain knowledge, there are no fixed methods to choose a prior distribution. The subjective selection of the prior distribution is often cited as a drawback of the Bayesian estimation process, as different priors lead to different results for the posterior distribution [81]. We set the prior distribution on $\theta_{X_i}$ for each node $X_i \in \mathcal{G}$ to follow a Dirichlet distribution. A Dirichlet prior distribution under a multinomial likelihood causes the posterior distribution to also follow a Dirichlet distribution. This is because the multinomial and Dirichlet distributions belong to conjugate families of distributions [82,83]. Therefore, we have the following formulation for the posterior distribution on $\theta_{X_i}$:

$$\theta_{X_i} \sim Dirichlet(\boldsymbol{\alpha}) \tag{5}$$

$$\boldsymbol{\alpha} = [\alpha_{s=1}, \alpha_{s=0}, \alpha_{s=-1}]$$

$$Dirichlet(\theta_{X_i}; \boldsymbol{\alpha}) = \frac{1}{\beta(\boldsymbol{\alpha})} \prod_{s \in \mathbf{S}} [\theta_{X_i=s}]^{\alpha_s - 1} \tag{6}$$

where $\beta(\boldsymbol{\alpha})$ is the Multivariate Beta function

$$P(\theta_{X_i}|X_i) = Dirichlet(\boldsymbol{\alpha}') \tag{7}$$

and

$$\boldsymbol{\alpha}' = [\alpha_{s=1} + M_{X_i}[s = 1], \alpha_{s=0} + M_{X_i}[s = 0], \alpha_{s=-1} + M_{X_i}[s = -1]]$$

$$\boldsymbol{\alpha}' = [\alpha'_{s=1}, \alpha'_{s=0}, \alpha'_{s=-1}]$$

In our study, we specifically set the prior distribution on each node $X_i$ to be Dirichlet ($\alpha_{s=1} = 1, \alpha_{s=0} = 1, \alpha_{s=-1} = 1$), which corresponds to uniform distribution over the open standard 2-simplex and is a noninformative prior distribution [84,85]. This is an appropriate choice for the prior distribution in our study, as we do not have prior knowledge regarding the distribution of each node in the BN. Furthermore, this assumption on the prior distribution of the nodes allows us to obtain a closed form solution for the posterior distribution. Selecting a different prior distribution will often lead to nonclosed form solution for the posterior distribtion and calculating the probability of data ($P(\mathcal{D})$) can be computationally expensive [86]. The formulation in Equation (7) represents the posterior distribution of the node parameter $\theta_{X_i}$. We approximate $\theta_{X_i}$ by its expected value in order

to obtain a point estimate for the LPDs in the BN. The expectation of a Dirichlet distribution is given by [87]:

$$
\theta_{X_i} = \begin{bmatrix} \theta_{X_i=1} \\ \theta_{X_i=0} \\ \theta_{X_i=-1} \end{bmatrix} \approx E[\theta_{X_i}|X_i] = \begin{bmatrix} \dfrac{\alpha'_{s=1}}{\sum_{\mathbf{S}} \alpha'_s} \\ \dfrac{\alpha'_{s=0}}{\sum_{\mathbf{S}} \alpha'_s} \\ \dfrac{\alpha'_{s=-1}}{\sum_{\mathbf{S}} \alpha'_s} \end{bmatrix} \tag{8}
$$

Similarly, if we have a node $X_i$ with a parent node $Y_i = s$ ($s \in \mathbf{S}$) under the same Dirichlet and multinomial framework, then the LPD associated with $\theta_{X_i|Y_i}$ can formulated as follows:

$$
\theta_{X_i|Y_i=s} = \begin{bmatrix} \theta_{X_i=1|Y_i=s} \\ \theta_{X_i=0|Y_i=s} \\ \theta_{X_i=-1|Y_i=s} \end{bmatrix} \approx E[\theta_{X_i|Y_i=s}|(X_i \mid Y_i = s)] = \begin{bmatrix} \dfrac{\alpha_{s=1}+M_{X_i|Y_i}[X_i=1,Y_i=s]}{\sum_{\mathbf{S}} \alpha_s+M_{X_i|Y_i}[X_i=1,Y_i=s]} \\ \dfrac{\alpha_{s=0}+M_{X_i|Y_i}[X_i=0,Y_i=s]}{\sum_{\mathbf{S}} \alpha_s+M_{X_i|Y_i}[X_i=0,Y_i=s]} \\ \dfrac{\alpha_{s=-1}+M_{X_i|Y_i}[X_i=-1,Y_i=s]}{\sum_{\mathbf{S}} \alpha_s+M_{X_i|Y_i}[X_i=-1,Y_i=s]} \end{bmatrix} \tag{9}
$$

In Equation (9), $M_{X_i|Y_i}[X_i = 1, Y_i = s]$ represents the frequencies when $X_i = 1$ and $Y_i = s$ simultaneously in the data set $\mathcal{D}$. Similarly, $M_{X_i|Y_i}[X_i = 0, Y_i = s]$ is the frequency of data points in $\mathcal{D}$ when $X_i = 0$ and $Y_i = s$ simultaneously, and so on for $X_i = -1$. Once the node parameters are estimated, gene intervention simulations can be carried out using inference in the BN. Inference computes the effect of intervening at each node on the reporter gene *LYSA* (gene N).

### 2.3. Gene Intervention Simulations

BNs represent the cause and effect relationship among the nodes of the system being modeled. Inference quantifies the cause and effect relationship by allowing us to compute conditional probability queries. Then, for a node of interest X, also known as the query node and an intervention (or evidence) node E in the BN, we can compute the conditional probability $P(X|E)$ using inference algorithms. This implies that we can calculate its effect on node X if we instantiate (fix) node E. Inference algorithms use the network parameters and structural dependencies to compute the required conditional probabilities. To further elucidate this notion, consider the BN shown in Figure 4. Let each node of the BN be a binary random variable with states 0 and 1. Suppose we have estimated the LPDs $P(A)$, $P(B|A)$, $P(C|A)$, and $P(D|B,C)$; then, we can use inference in this BN to answer conditional probability queries such as $P(D = 1|A = 1)$.

We compute $P(D = 1|A = 1)$ as follows:

$$
P(D = 1|A = 1) = \frac{P(D = 1, A = 1)}{P(A = 1)}
$$
$$
= \frac{\sum_B \sum_C P(A = 1, B, C, D = 1)}{P(A = 1)}
$$

Using the properties of the BN, all nodes are independent of any nondescendant nodes

$$
P(D = 1|A = 1) = \frac{\sum_B \sum_C P(A = 1)P(B|A)P(C|A = 1)P(D = 1|B,C)}{P(A = 1)}
$$
$$
= \sum_B \sum_C P(B|A = 1)P(C|A = 1)P(D = 1|B,C)
$$

We can use the LPDs to calculate the exact probability $P(D = 1|A = 1)$.

Inference techniques such as the one applied in the BN in Figure 4 are classified as "exact" because they compute the true values for the conditional probability query. However, exact inference in BNs has been shown to be NP-hard [88,89]. While there exist efficient algorithms for exact inference, they are often limited to simpler DAG structures [88]. For example, Pearl's message-passing algorithm works efficiently for singly connected DAG structures [90]. Therefore, for larger DAGs, exact inference is not ideal as the computational cost of calculating the conditional probabilities can be expensive. In such cases, we employ approximate inference algorithms, which produce estimates of the exact conditional probabilities [91]. Approximate inference can include wide-ranging techniques such as model simplification methods, loopy belief propagation methods, search-based methods, utility-based methods, and stochastic simulation methods [92]. In this paper, we implement a stochastic simulation-based inference technique called likelihood weighting (LW) to estimate the conditional probability queries in the BN model for the lysine biosynthesis pathway. Stochastic simulation techniques estimate the conditional probabilities by drawing samples from the LPDs. These estimates typically converge to the true conditional probabilities as the number of samples drawn increases. LW can efficiently handle inference of large multiply connected BNs and is based on forward sampling [92,93]. Since our BN model is multiply connected and we are only interested in estimating $P(N = 1 \mid E \in \{A, B, C, ..., M\})$, i.e., the probability of upregulating *LYSA* (gene N), while conditioning on other genes (evidence or intervention nodes), LW turns out be a suitable method for performing inference.

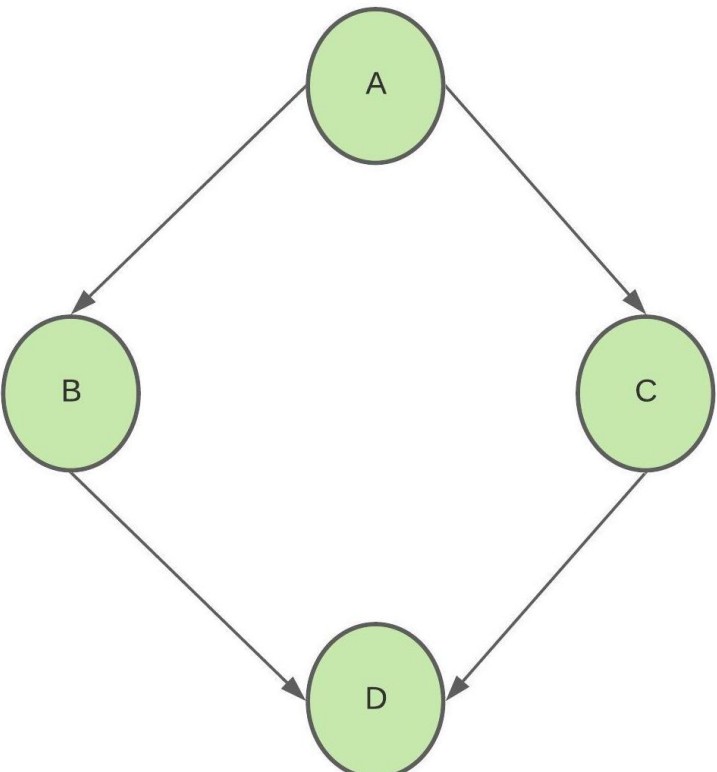

**Figure 4.** Example BN with binary nodes.

The LW algorithm estimates the conditional probability, $P(X = x \mid E = e)$ for a query node X and an evidence node E, by generating samples from a BN model. We fix the sample size (*m*) and a topological ordering at the start of the algorithm. The algorithm iterates through a sample generation process *m* times, and then computes the conditional probability from the generated samples. During the sample generation process, the algorithm generates values for the nonevidence nodes only; it sets the value of the evidence node to its observed (*e*, in this case) value. The node values for each sample are

generated in the established topological ordering. Each sample is assigned a weight of 1 at the start of the sample generation process. The weight is updated only when an evidence node is encountered while traversing the topological ordering. When this happens, the sample's weight is updated by multiplying the current weight with the likelihood of the evidence node conditioned on the state of its parent nodes. The likelihood is given by the probability $P(E = e \mid P_a(E))$. The process is repeated until $m$ samples are generated. Following this step, conditional probability is estimated by dividing the sum of the weights of the samples where $X = x$ by the sum of all the sample's weights. The pseudocode for the LW algorithm by Stuart Russell and Peter Norvig is presented in Algorithm 1 [94].

---

**Algorithm 1:** Likelihood-Weighting Algorithm

---

**Function** LIKELIHOOD-WEIGHTING(*X, e, bn,N*):
    **outputs** an estimate of P(X|e)
    **inputs:** X, the query variable
             **e**, observed values for variables **E**
             bn, a Bayesian network specifying joint distribution $P(X_1,..,X_n)$
             N, the total number of samples to be generated
    **local variables:W,** a vector of weighted counts for each value of X, initially zero
    **for** *j=1 to N* **do**
        **x**,w ← WEIGHTED-SAMPLE(bn,**e**)
        **W**[x] ← **W**[x] + w where x is the value of X in **x**
    **end**
    **return** NORMALIZE(**W**)

**Function** WEIGHTED-SAMPLE(*bn, **e***):
    **outputs** an event and a weight
    w ← 1; **x** ← an event with n elements initialized from **e**
    **for** *each* variable $X_i$ *in* $X_1,..,X_n$ **do**
        **if** $X_i$ *is an evidence variable with value* $X_i$ *in **e*** **then**
            w ← w × P($X_i = X_i$ | parents($_i$))
        **else**
            **x**[i]← a random sample from P($X_i$ | parents($X_i$))
        **end**
    **end**
    **return x**,w

---

### 2.4. Data Set

To estimate the LPDs in the BN model, we use the data set GSE98455, which is publicly available from the NCBI GEO database [95–97]. This data set was selected because it contains RNA-Seq counts for rice seedlings under saline stress and normal (unstressed or control) conditions and had the highest number of samples (data points) per gene available among publicly available data sets. The entire data set contained 57,846 rows (genes) and 368 columns (control and saline stress). Since our BN model contains nodes modeled as categorical variables, the RNA-Seq data had to be preprocessed. The data preprocessing steps are outlined as follows:

1. The entire data set was normalized using the ratio of medians methods.
2. We selected the data for the genes A–N, as these were the genes in the BN model. We identified the data for each of the genes by mapping their data set IDs to their respective MSU IDs. This reduced our data set to a size of 14 rows (Gene A–N) and 368 columns.
3. We further segregated the normalized data set based on saline stress and normal conditions. Since the number of columns for saline stress and normal conditions were the same, each of the resulting data set had 14 rows and 184 columns.

4. We ran *K*-means clustering separately on both the saline stress and normal conditions data set to convert them from normalized to categorical values. The clustering process categorized the data in both the data sets into the following values 1 (active), 0 (dormant), and −1 (inhibited). The low expression values were categorized to the value of −1, the high expression values were categorized to the value of 1, and the remaining expression values in the middle were categorized to a value of 0.

Once the categorical values were obtained for both the treatment and control data sets, the LPDs were estimated under each case using the Bayesian approach described in the *Parameter Estimation* section. We then ran LW to simulate gene intervention. The ratio of medians methods used for normalization is described in the DESeq2 data processing protocols by Love et al. [98]. DESeq2 is one of the most commonly used RNA-Seq data processing protocols and is easily accessible in the R programming language as a package (DESeq2) [99–102]. The file for mapping data set IDs to MSU IDs was provided to us by the authors of the data set GSE98455. We have highlighted their contribution in the acknowledgment section. A visual representation of the data processing pipeline is presented in Figure 5. Figures 6 and 7 show the discretized categorical data for each node in the BN under normal and saline stress conditions, respectively.

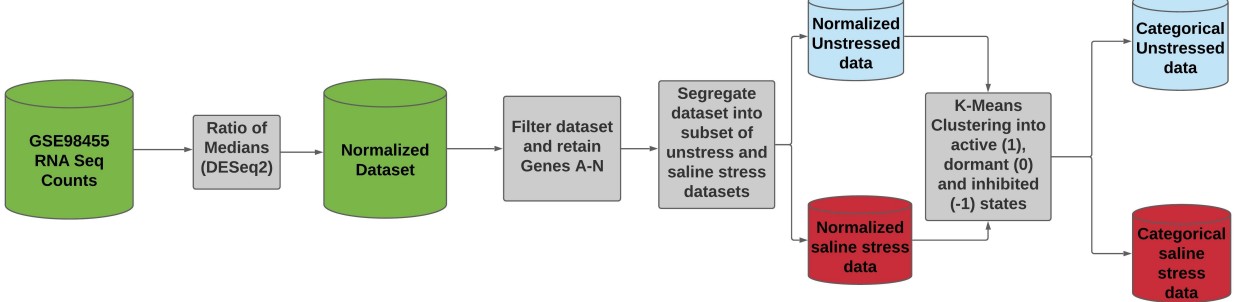

**Figure 5.** Data processing pipeline for RNA-Seq data set GSE98455.

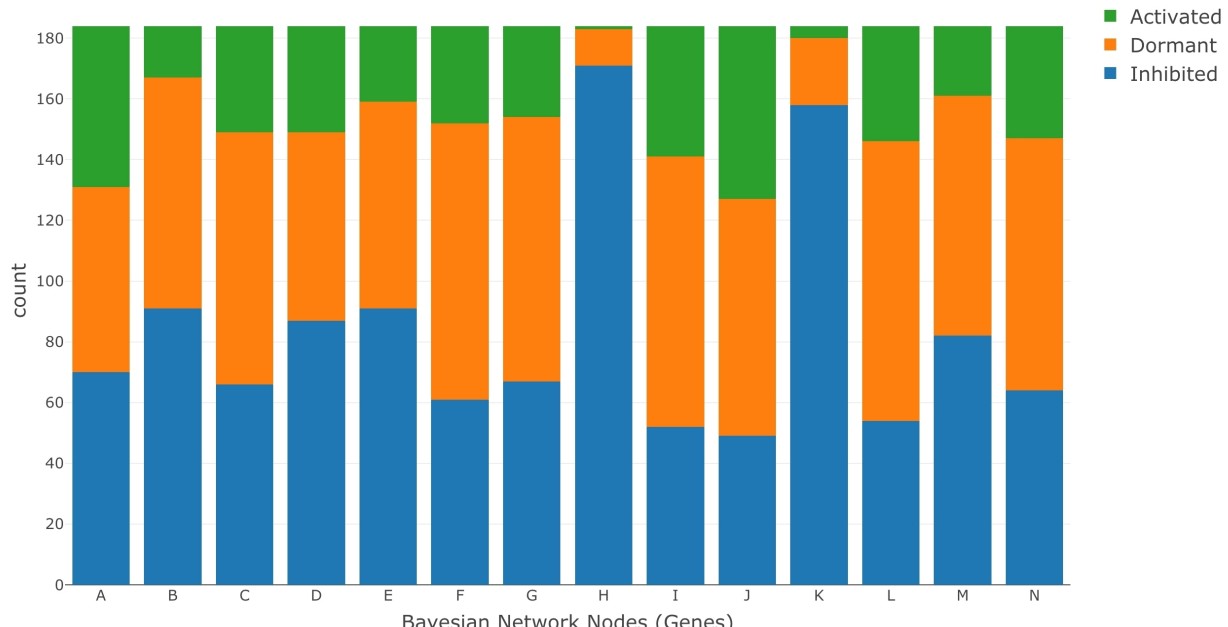

**Figure 6.** Discretized RNA-Seq data under normal conditions.

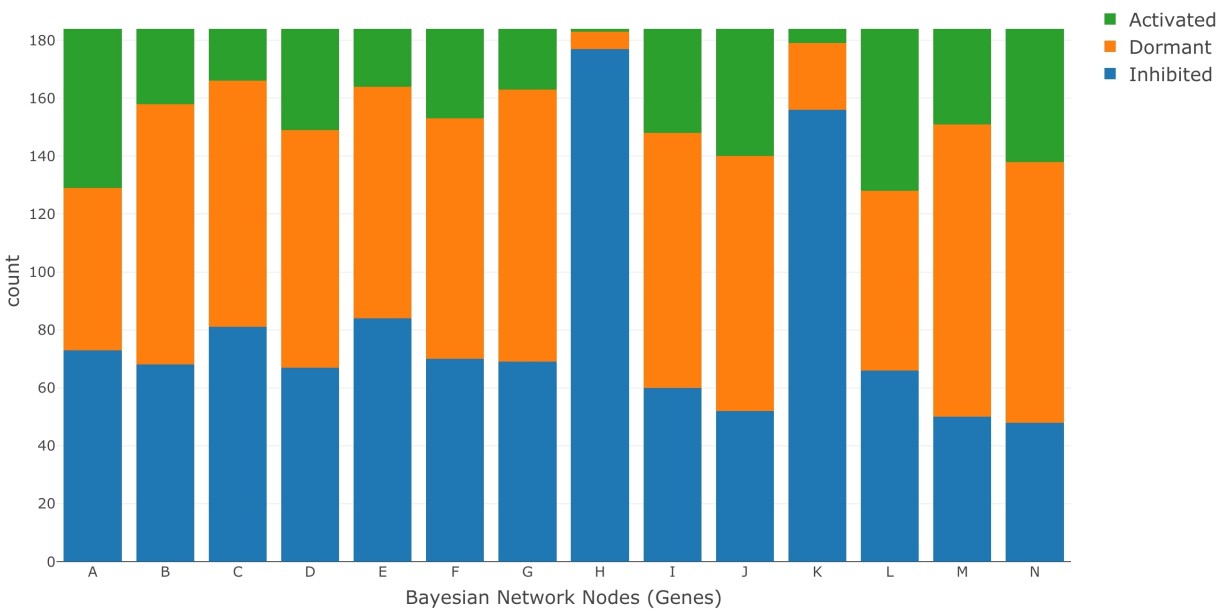

**Figure 7.** Discretized RNA-Seq data under saline stress conditions.

### 3. Results

The LPDs estimated from the RNA-Seq data set were used to simulate gene intervention in the BN. When intervening at a gene, the node representing that gene in the BN was instantiated to a status of active (1), dormant (0), or inhibited ($-1$). We applied the LW algorithm with a large sample size of 600,000 to compute the probability $P(N = 1 \mid$ gene intervention) and ensure convergence of the probabilities being estimated. Gene N (*LYSA*) is set as the query node because it is the reporter gene for lysine production; thus, upregulating gene N (*LYSA*) may lead to increased lysine production. We perform intervention at genes A–M one at a time and then in combinations of two (pairs) at a time. These gene intervention strategies were applied under both normal and saline stress conditions. In order to measure the causal effect of intervention, we subtract the marginal probability $P(N = 1)$ from $P(N = 1 \mid$ gene intervention), for all the possible gene intervention strategies. This difference is defined as the score metric and is used to compare the effectiveness of each gene intervention strategy. The data processing and probability computation pipeline was written in the R programming language, and the Bnlearn package was used to perform LW [103–105]. So,

$$score = P(N = 1 \mid \text{gene intervention}) - P(N = 1). \tag{10}$$

Since there are many possible combinations under single and pairwise gene interventions, we only include the top five intervention strategies with the highest scores in Figure 8 and Tables 1 and 2. The proteins encoded by each of the genes in these intervention strategies are summarized in Table 3. In Figure 8a,b, we present the scores for single node intervention under normal and saline stress conditions.

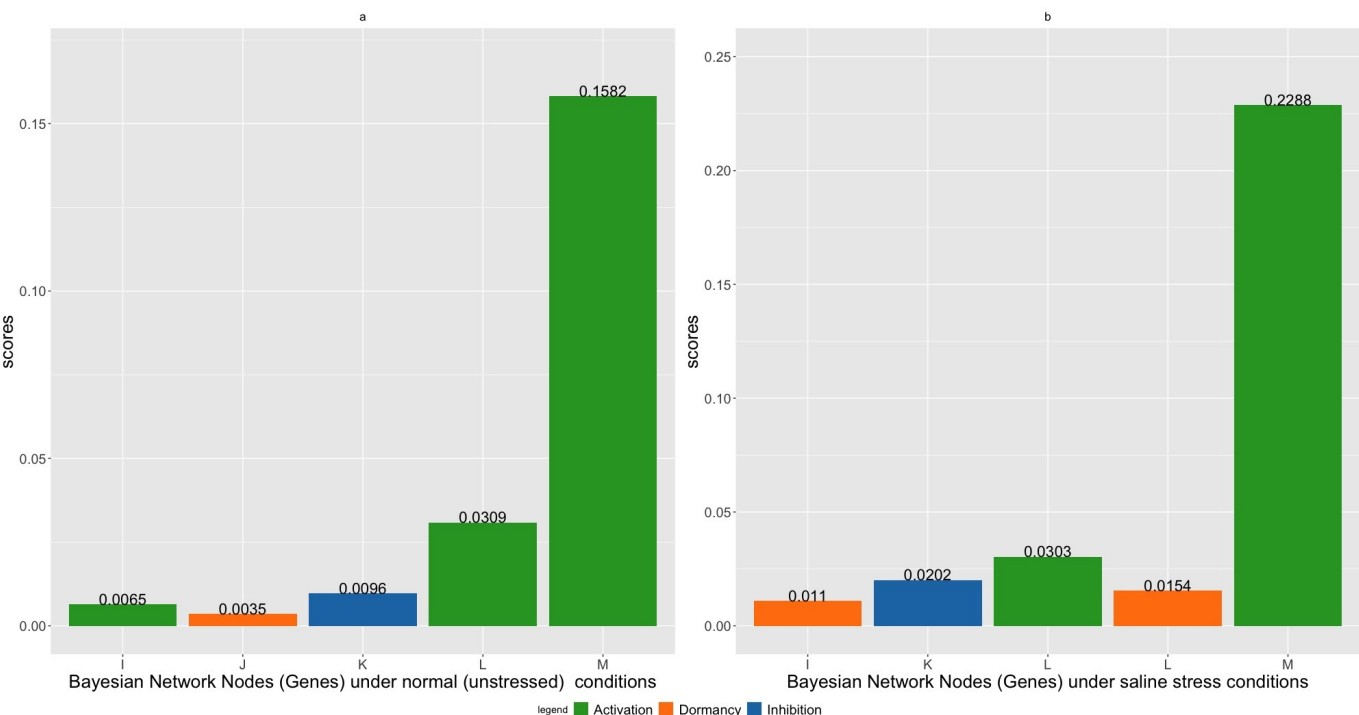

**Figure 8.** Single node intervention under (**a**) normal and (**b**) saline stress conditions.

It is clear from Figure 8a,b that activating gene M (*DAPF*) has the maximum score. This implies that, under both normal and saline stress conditions, genetically activating gene M (*DAPF*) has the best chance for upregulating the reporter gene N (*LYSA*). We also notice that gene L (*AGD2*) is also fairly active in its role in upregulating gene N (*LYSA*). Activating gene L (*AGD2*) achieves the second largest score under normal conditions. Under saline stress conditions, activating gene L (*AGD2*) or keeping it dormant also ranks among the top five gene intervention strategies. Inhibiting gene K (*ALD1*) achieves the third and second highest scores under normal and saline stress conditions. Additionally, we also observe that midstream genes such as gene I (*DAPB1*) and gene J (*DAPB2*) also play an active role in upregulating gene N (*LYSA*). However, activating gene M (*DAPF*) has a significantly higher score under both conditions; thus, gene M (*DAPF*) servers as an ideal candidate for gene intervention.

**Table 1.** Top five pairwise intervention strategies under normal conditions.

| Index | Gene Name/Alias | Intervention | Gene Name/Alias | Intervention | Score |
|-------|-----------------|--------------|-----------------|--------------|-------|
| 1 | *ALD1* (Gene K) | Active | *DAPF* (Gene M) | Active | 0.1657 |
| 2 | *ALD1* (Gene K) | Dormant | *DAPF*(Gene M) | Active | 0.1653 |
| 3 | *AGD2* (Gene L) | Inhibited | *DAPF* (Gene M) | Active | 0.1639 |
| 4 | Gene A | Active | *DAPF* (Gene M) | Active | 0.1637 |
| 5 | Gene C | Active | *DAPF* (Gene M) | Active | 0.1634 |

**Table 2.** Top five pairwise intervention strategies under saline stress conditions.

| Index | Gene Name/Alias | Intervention | Gene Name/Alias | Intervention | Score |
|-------|-----------------|--------------|-----------------|--------------|-------|
| 1 | Gene F | Dormant | *DAPF* (Gene M) | Active | 0.2322 |
| 2 | *ALD1* (Gene K) | Dormant | *DAPF*(Gene M) | Active | 0.2321 |
| 3 | Gene B | Inhibited | *DAPF* (Gene M) | Active | 0.2312 |
| 4 | Gene E | Inhibited | *DAPF* (Gene M) | Active | 0.2306 |
| 5 | Gene A | Dormant | *DAPF* (Gene M) | Active | 0.2305 |

Tables 1 and 2 represent the five highest-scoring pairwise intervention strategies for normal and saline stress conditions. Each table contains gene names or alias along with their intervention strategies. The tables are arranged in descending order of the score. Under each condition, the score amongst different five highest-scoring strategies are almost similar with marginal differences. Under normal conditions in Table 1, we observe that activating both gene K (*ALD1*) and gene M (*DAPF*) maximized the scores. While under saline stress, keeping gene F (LOC_Os03g55280) dormant and activating gene M (*DAPF*) achieved the highest score. This implies that under each of the conditions, the respective pairwise intervention strategy with highest scores maximize the likelihood of upregulating gene N (*LYSA*). From Tables 1 and 2, it can also be seen that upstream genes such as genes A, B, C, and E are also involved in the upregulation of gene N (*LYSA*) and produce comparable scores to those produced by the regulation of downstream genes such as gene K (*ALD1*) and gene L (*AGD2*). Across both the conditions, we also observe that gene M (*DAPF*) is always upregulated, which serves to be a further indicator of the high regulatory effect of gene M (*DAPF*) on gene N (*LYSA*). We should note that these rankings in Tables 1 and 2 might vary slightly upon rerunning the simulations, as LW is based on a stochastic simulation process, which may cause minor variation in estimating the probabilities required for computing the score metric. However, this does not affect our overarching conclusion that *DAPF* is the most potent regulator of *LYSA*, as it is present and upregulated in all the top five strategies under pairwise intervention. Furthermore, under single intervention, *DAPF* scores significantly higher than rest of the genes.

**Table 3.** Protein encoded by intervention genes in Figure 8 and Tables 1 and 2.

| Gene Alias/Name | MSU IDs | Protein |
|---|---|---|
| Gene A | LOC_Os01g70300 | Aspartokinase 3, chloroplast precursor, putative, expressed |
| Gene B | LOC_Os03g63330 | Aspartokinase, chloroplast precursor, putative, expressed |
| Gene C | LOC_Os07g20544 | Aspartokinase, chloroplast precursor, putative, expressed |
| Gene E | LOC_Os09g12290 | Bifunctional aspartokinase/homoserine dehydrogenase, chloroplast precursor, putative, expressed |
| Gene F | LOC_Os03g55280 | Semialdehyde dehydrogenase, NAD binding domain containing protein, putative, expressed |
| Gene I/*DAPB1* | LOC_Os02g24020 | Dihydrodipicolinate reductase, putative, expressed |
| Gene J/ *DAPB2* | LOC_Os03g14120 | Dihydrodipicolinate reductase, putative, expressed |
| Gene K/*ALD1* | LOC_Os03g09910 | Aminotransferase, classes I and II, domain containing protein, expressed |
| Gene L/*AGD2* | LOC_Os03g18810 | Aminotransferase, classes I and II, domain containing protein, expressed |
| Gene M/*DAPF* | LOC_Os12g37960 | Diaminopimelate epimerase, chloroplast precursor, putative, expressed |

## 4. Discussion

In this paper, we studied the lysine biosynthesis pathway in rice to identify the genetic regulators of lysine content. Rice is a staple food source for 50% of the global population; with lysine being the first limiting essential amino acid in rice, it is vital to identify gene regulators that can boost lysine content. We modeled the lysine biosynthesis pathway in rice using BNs under normal and saline stress conditions to identify these regulators. We used BNs because they allow us to integrate domain knowledge in the form of pathway information with experimental data. We used publicly available RNA-Seq data to estimate the LPDs in the BN and run the gene intervention simulations. We intervened at the genes one at a time and then in pairwise combinations using the LW inference algorithm. We calculated a score metric to measure the efficacy of the gene intervention strategies.

Our analysis revealed that upregulating *DAPF* (gene M) maximized the probability of the lysine reporter gene *LYSA* (gene N) being upregulated under both normal and saline stress conditions. When *DAPF* (gene M) was upregulated, it not only achieved the highest score under single gene intervention but was also present in all the five highest-scoring gene intervention strategies under pairwise intervention. This implies that *DAPF* (gene M) is a positive regulator of *LYSA* (gene N) and serves as an ideal candidate for genetic intervention. Gene editing can be used to target and upregulate *DAPF* (gene M) in rice. Wet lab experiments involving *DAPF* overexpressing rice can confirm if this intervention strategy upregulates *LYSA* and increases the overall lysine content. We further

observed under single gene intervention that midstream genes such as *DAPB1* (gene I) and *DAPB2* (gene J) also played significant roles in upregulating *LYSA* (gene N). Under pairwise intervention, we found that upstream genes such as genes A, B, C, and E were also involved in upregulating *LYSA* (gene N).

Future steps in our study of lysine include confirming our finding in this paper by performing wet lab experiments. We would also like to improve our choice of the prior distribution on each node. In our current analysis, we used a noninformative prior distribution, as we did not have any knowledge regarding the prior distribution of the nodes in the BN. Using informative prior distributions may increase the computational costs, but it has the potential to improve our predictions of lysine regulators. Furthermore, we are also interested in studying how other essential amino acids such as threonine, methionine, and isoleucine in the larger aspartate pathway regulate lysine content. Threonine is known to downregulate the enzyme AK in the lysine biosynthesis pathway; thus, studying the multilevel regulation among the different amino acids in the aspartate pathway will help deepen our understanding of lysine production.

## 5. Conclusions

We modeled the lysine biosynthesis pathway in rice under normal and saline stress conditions to identify the regulators of lysine. Among the essential amino acids, lysine is present in the least quantity in rice; thus, increasing its content in rice will improve its nutritional value. Our analysis revealed that, under both the normal and saline stress conditions, upregulating *DAPF* is the best genetic intervention strategy for upregulating the lysine reporter gene *LYSA*. Applying gene intervention techniques such as CRISPR-Cas9-based gene editing to upregulate *DAPF* has the potential to increase the lysine content in rice.

**Supplementary Materials:** The following are available online at https://www.mdpi.com/article/10.3390/inventions6020037/s1.

**Author Contributions:** Conceptualization, E.M.S. and A.D.; methodology, A.L. and K.R; software, A.L.; validation, E.M.S. and K.R.; formal analysis, A.L. and A.D.; investigation, A.L. and A.D.; resources, A.D. and E.M.S.; data curation, A.L.; writing—original draft preparation, A.L.; writing—review and editing, A.L., A.D., K.R., and E.M.S.; visualization, A.L.; supervision, E.M.S. and A.D.; project administration, E.M.S. and A.D.; funding acquisition, E.M.S. and A.D. All authors have read and agreed to the published version of the manuscript.

**Funding:** This work was supported in part by the TEES-AgriLife Center for Bioinformatics and Genomic Systems Engineering (CBGSE) startup funds, the Texas A&M X-Grant Program, and in part by the National Science Foundation under Grant ECCS-1609236 (to A.D.). The funding bodies did not play any role in the design of the study and collection, analysis, and interpretation of data and in writing the manuscript.

**Institutional Review Board Statement:** Not applicable.

**Informed Consent Statement:** Not applicable.

**Data Availability Statement:** The data sets used in this study are publicly available at the NCBI with the accession numbers of GSE98455. The subset of data extracted from the data set to support the conclusion of this article are included within the article. The R code containing the simulations are provided as supplemental files. The R code files are also publicly avaialble at the following Github repository: https://github.com/adilahiri/Lysine_Regulators (accessed on 16 May 2021).

**Acknowledgments:** We would like to recognize Chi Zhang, Associate Professor, School of Biological Sciences, University of Nebraska, Lincoln. He generously provided us with the gene annotation file required for analyzing the data set GSE98455.

**Conflicts of Interest:** The authors declare no conflict of interest.

## Abbreviations

The following abbreviations are used in this manuscript:

LKR     Lysine ketoglutarate reductase
SDH     Saccharopine dehydrogenase
DHPS    Dihydrodipicolinate synthase
AK      Aspartate kinase
GRN     Gene regulatory network
GMO     Genetically modified organisms
MSU     Michigan State University
TF      Transcription factor
BN      Bayesian network
PGM     Probabilistic graphical model
LPD     Local probability distribution
i.i.d   Independent and identically distributed
LW      Likelihood weighting

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
