# Peer review of "Bayesian Network Analysis of Lysine Biosynthesis Pathway in Rice"

_inventions, doi:10.3390/inventions6020037_

Round 1

Reviewer 1 Report

Please see attachment, thank you.

Author Response

We thank you very much for reviewing our manuscript. We appreciate your time and effort in providing us with constructive feedback towards improving our manuscript. 

Reviewer 2 Report

This interdisciplinary work can be of interest for the biotechnological application aimed at increasing crop nutritional value as described by the authors.

However, in my opinion, some important information relative to the biological context are missing.

In the introduction section, the authors should add a picture reporting the biosynthetic route of lysine. In the same section, please report the effect of high soil salinity on lysine level in rice.

The authors should add information about the proteins encoded by the reported genes. This is also important for supporting conclusions: it is not possible to suggest a gene as a putative target for biotechnological manipulation without knowing the biological context in which the correspondent protein is involved.

It is not clear how data from GEO database have been selected. Did the authors consider the uniformity of growth conditions and developmental stage of rice seedlings and/or the intensity of salt stress (i.e. NaCl concentration)? Indeed, different experimental conditions could give opposite effects on gene expression and metabolism.

Author Response

(The authors gave the same response as above.)

Round 2

Reviewer 1 Report

I thanks reviewers very much to take my comments and provided their point by point responses for each of my question.
In this revision, the authors made substantial change on the manuscript to make significant improvement.
The authors enhanced the quality of presentation of other Figures and also included a new Figure to explain lysine metabolism.
I think all my concerns were well addressed in this version. Thank you